# Blind Deblurring Based on Sigmoid Function

**DOI:** 10.3390/s21103484

**Published:** 2021-05-17

**Authors:** Shuhan Sun, Lizhen Duan, Zhiyong Xu, Jianlin Zhang

**Affiliations:** 1Key Laboratory of Optical Engineering, Chinese Academy of Sciences, Chengdu 610209, China; sunshuhan17@mails.ucas.ac.cn (S.S.); duanlizhen18@mails.ucas.ac.cn (L.D.); xzy158@163.com (Z.X.); 2School of Electronic, Electrical and Communication Engineering, University of Chinese Academy of Sciences, Beijing 100049, China; 3Institute of Optics and Electronics, Chinese Academy of Sciences, Chengdu 610209, China

**Keywords:** image processing, blind deblurring, image deblurring, inverse problem

## Abstract

Blind image deblurring, also known as blind image deconvolution, is a long-standing challenge in the field of image processing and low-level vision. To restore a clear version of a severely degraded image, this paper proposes a blind deblurring algorithm based on the sigmoid function, which constructs novel blind deblurring estimators for both the original image and the degradation process by exploring the excellent property of sigmoid function and considering image derivative constraints. Owing to these symmetric and non-linear estimators of low computation complexity, high-quality images can be obtained by the algorithm. The algorithm is also extended to image sequences. The sigmoid function enables the proposed algorithm to achieve state-of-the-art performance in various scenarios, including natural, text, face, and low-illumination images. Furthermore, the method can be extended naturally to non-uniform deblurring. Quantitative and qualitative experimental evaluations indicate that the algorithm can remove the blur effect and improve the image quality of actual and simulated images. Finally, the use of sigmoid function provides a new approach to algorithm performance optimization in the field of image restoration.

## 1. Introduction

Digital images are an important source of information for humans. However, due to the imaging equipment’s defects (optical aberration, defocusing, etc.) and limitations of shooting conditions (insufficient light, bad weather, and atmosphere turbulence), images obtained will be of low visual quality. It is a blind deconvolution problem which calls for a solution to recover the scene or restore the clear picture from its blurred counterparts with unknown blur parameters. Blind deconvolution is a well-known, ill-posed problem. This paper also takes the effects of noise into account. To obtain an image of high visual quality, it is necessary to strike a balance between resolution and noise suppression. In the deblurring image discussion, the obtained blurred vision g(x,y) is modeled as the convolution between a clear image o(x,y) and the point spread function (PSF) h(x,y) and the additive noise n(x,y). The PSF, also known as blur kernel [1], causes image degradation. In the image restoration literature, image degradation is commonly modeled as follows [2]
(1)g(x,y)=o(x,y)∗h(x,y)+n(x,y)
where “*” is the convolution operator, o(x,y) and g(x,y) stand for the given clear image and its degraded counterpart, respectively, h(x,y) denotes the point spread function (PSF) representing degradation induced in the spatial domain, and n(x,y) represents the additive noise.

Image blurring is a significant detriment to the succeeding work, such as object recognition and object tracking. Therefore, image restoration technology has attracted extensive attention. Many academics have presented meaningful work. Categorizing by problem-solving approach, there are four main types of image restoration method. They are image restoration algorithms in the spatial domain, image restoration algorithm in the frequency domain, image modeling and image estimation algorithm, and neural network algorithm. The most common type of image restoration algorithms is those in the spatial domain. This algorithm was proposed and used first. The most representative algorithm in the spatial domain is the regularization method [3]. For an ill-posed problem, the condition number is considerable. By adding regularization to the loss function, the approach can use the original image’s priors to reduce the condition number. The approach can obtain a good result quickly for image restoration algorithms in the frequency domain [4]. The approach maps the different frequency characteristics of the image according to the flat region and the edge region. The approach converts the image to the frequency domain through the transformation model [5,6]. After completing data processing in the frequency domain, it converts the results to the spatial domain. The filtering method is widely used. The most typical method is Wiener filtering, which essentially minimizes the mean square error (MSE). The study of stochastic processes is always a hot topic. Gaussian random field theory and Markov field theory are well known. They apply Bayesian theory to image restoration. The most critical probabilistic models are maximum likelihood estimation (MLE) and maximum a posterior estimation (MAP) [7], by which the image restoration problem is converted to probability estimation through Bayesian inference. Maximum Likelihood Estimation algorithm and the Richardson–Lucy algorithm (RL) [8] are the most representative. A multiplicative iterative approach (MIA) [9,10] was proposed based on a probabilistic model. MIA [9,10] naturally preserves the non-negative constraint on the iterative solutions when the initial estimates are non-negative, producing a restored image of high quality. At present, neural networks are the most popular in computer vision. An artificial neural network [11,12] is a new method to obtain the loss function’s minimum value. However, artificial neural networks tend to be more expensive in terms of computation complexity.

When restoring image sequences, it is usually assumed that the image sequences’ target does not change significantly in a short time. The estimation of adjacent frames of short exposure sequence is applied to approximate the current frame and to obtain a better target image estimation. Due to the redundancy of information, image sequences provide more supplementary information for image recovery. Compared with single image restoration, image sequences can reduce the meaningless solution and improve restoration stability. Unfortunately, there are two sides to everything, like a coin. Image sequences always need more storage memory. Additionally, the need for image information of adjacent frames leads to more computation.

In this paper, an efficient scheme for blind deblurring is introduced via the sigmoid function, which was inspired by the multiplicative iterative algorithm (MIA) [9,10]. The MIA, as reported in [9,10], is efficient but limited to weak degradation. To overcome this drawback and deal with the severe degradation problem, a new form of iteration strategy is adopted in this work, which employs the sigmoid function, leading to a novel blind deconvolution algorithm for restoration of seriously degraded and blurred images.

The contributions are as follows. First, this paper proposes an image restoration model based on the sigmoid function. Intuitively, the latest iteration model can ensure that the image is non-negative in the iteration process. As a result, it does not need any other constraints to make the pixel value non-negative. Second, the approach can effectively restore severely degraded images using the sigmoid function and the information between sequences. Compared with the classical and the state-of-the-art methods, experiments show that the new method has a better competitive performance for severely degraded images. Third, to better evaluate the algorithm’s performance, this paper presents more experimental results of blind deblurring. The results demonstrate that the new algorithm can achieve the same level of performance as state-of-the-art methods.

## 2. Related Work

In recent years, significant progress has been made in image deblurring [13]. In particular, using the prior information of the image to deblur has attracted significant attention from academics. Many contributions reported in the literature are based on maximum a posterior (MAP) framework and variational Bayesian methods [14,15,16,17,18,19,20,21,22,23,24,25,26,27,28,29,30,31,32]. These methods often involve two steps. In the first step, the blur kernel is estimated by using the obtained image. The second step is to estimate the latent image according to the estimated blur kernel through a non-blind deconvolution method [33,34,35,36]. Considering that the simplest MAP method cannot always estimate the blur kernel effectively, it is not easy to obtain a satisfactory image.

The key to image deblurring is to use the image’s prior information to constrain the blur kernel and the image. The most widely used prior is gradient sparsity prior [37,38,39,40]. However, in reference [18], the authors find that the gradient sparsity prior is often more friendly to blurred images than to clear images. In reference [14,15,16,29,41,42], the sharp edges of the image are constrained in order to alleviate the above shortcomings. However, it has to be acknowledged that images do not always have sharp edges; for example, many natural images have unsharp edges. At the same time, some other image priors are also widely used by scholars. For example, intensity prior [22], normalized sparsity prior [19], dark channel prior [23], data-driven learned prior [28]. These image priors have also achieved remarkable results.

With the popularity of deep neural networks, data-driven methods have also achieved great success [43,44,45,46,47,48]. In reference [43], Sun et al. adopted a convolutional neural network (CNN) to remove motion blur. Nah et al. [45] designed a multi-scale convolutional neural network that can restore the image without estimating the blur kernel. Furthermore, Kupyn et al. [46] designed a generative adversarial network (GAN) to restore images end-to-end. Su et al. [47], applied an improved convolutional neural network to video deblurring. Yang et al. [47] designed a 3D convolutional encoder–decoder network for video deblurring. The data-driven methods do not always generalize well if the test images vary or differ from the training dataset.

Having reviewed image restoration progress of the last decade in this section, the remaining contents of this article are organized as follows. In Section 3, a new blind deblurring algorithm based on the sigmoid function (BDA-SF for short) is introduced in detail with practical applications. In Section 4, experimental results are presented for performance evaluation, which are compared with those of the existing algorithms. Section 5 provides a summary of this paper.

## 3. Methods

### 3.1. Image Restoration Model

Based on the idea of the multiplicative iterative algorithm (MIA), which is efficient but limited to weak degradation, a novel blind deconvolution algorithm is devised employing the sigmoid function, i.e., the BDA-SF, for the restoration of seriously degraded images, to overcome MIA’s limitation. The algorithm has good convergence with simple parameter selection. The algorithm can avoid the instability of numerical calculation and naturally meet non-negative constraints. It has been shown that the performance of the least-squares algorithm is almost insensitive to whether noise is Poissonian or Gaussian [49], and that, for Poissonian noise, no strong difference exists between the results of the ISRA and those of the RLA, while for Gaussian noise, the ISRA produces much better results than the RLA [50]. Here, owing to the robustness of Gaussian noise hypothesis, the likelihood probability function [51] can be established as
(2)$$P\left(g|o,h\right)=\prod_{x,y}\frac{1}{\sqrt{2\pi }\sigma }exp\left(-\frac{{\left[g\left(x,y\right)-h\left(x,y\right)*o\left(x,y\right)\right]}^{2}}{2\sigma^{2}}\right)$$

The $\sigma^{2}$ is the variance in the noise, g(x,y) is the blurred image, o(x,y) is the original image, and h(x,y) is the point spread function (PSF). The corresponding log-likelihood [51] multiplied by $\sigma^{2}$ is
(3)$$\sigma^{2}log\left[P\left(g|o,h\right)\right]=\sum_{x,y}\sigma^{2}log\left[\frac{1}{\sqrt{2\pi }\sigma }\right]-\sum_{x,y}\frac{{\left[g\left(x,y\right)-h\left(x,y\right)*o\left(x,y\right)\right]}^{2}}{2}$$
(4)$$\begin{array}{cc} J(o,h) & =-\sigma^{2}log\left[P\left(g|o,h\right)\right]\\ & =\sum_{x,y}\frac{{\left[g\left(x,y\right)-h\left(x,y\right)*o\left(x,y\right)\right]}^{2}}{2}+C\\ \; & ={∥g\left(x,y\right)-h\left(x,y\right)*o\left(x,y\right)∥}^{2}+C \end{array}$$
where *C* is a constant independent of o(x,y) and h(x,y), J(o,h) is the loss function. Basically, the problem is highly ill-posed, and there are many different solution pairs (o,h) that give rise to the same *g* [22]. In order to make the problem well-posed, this paper uses sparsity prior to constrain the image and the kernel [20]. This paper uses ${∥h∥}_{1}$ instead of ${∥h∥}_{2}$ used in [20], which works to constrain the kernel to be sharp [17,52].
(5)$$p\left(o\right)={\alpha ∥\nabla o∥}_{0}$$
(6)$$p\left(h\right)={\gamma ∥h∥}_{1}$$
(7)p(o,h)=p(o)+p(h)
where α, γ are penalty parameters, “∇” is the gradient operator, $L_{0}$ norm is modeled by a numerical approximation function in [53], i.e., ${∥\nabla o∥}_{0}∼\frac{{∥\nabla o∥}_{2}^{2}}{{∥\nabla o∥}_{2}^{2}+\beta }$, where β is a modulation parameter (in this paper, β is set to 0.001). The loss function can be written as
(8)$$J\left(o,h\right)={∥g\left(x,y\right)-h\left(x,y\right)*o\left(x,y\right)∥}^{2}+p\left(o,h\right)$$

Just as the MIA [10], blind deconvolution is to minimize the loss function by obtain partial derivatives of J(o,h) with respect to o(x,y) and h(x,y), respectively, as follows
(9)$$\frac{\partial J(o,h)}{\partial o}=h^{c}\left(x,y\right)*\left[g\left(x,y\right)-h\left(x,y\right)*o\left(x,y\right)\right]+\nabla_{o}p\left(o,h\right)$$
(10)$$\frac{\partial J(o,h)}{\partial h}=o^{c}\left(x,y\right)*\left[g\left(x,y\right)-h\left(x,y\right)*o\left(x,y\right)\right]+\nabla_{h}p\left(o,h\right)$$
where the function $f^{c}\left(\right)$ represents the adjoint function of f(), $\nabla_{o}p\left(o,h\right)=-\alpha \nabla \cdot \frac{2\beta \nabla o}{{∥∥\nabla o∥}^{2}{+\beta ∥}_{2}^{2}}$, $\nabla_{h}p\left(o,h\right)=-\gamma \nabla \cdot \frac{\nabla h}{{∥\nabla h∥}_{2}}$. Forcing (Equation 9) and (Equation 10) to zero, it will arrive at the maximum log-likelihood equations:(11)$$h^{c}\left(x,y\right)*\left[g\left(x,y\right)-h\left(x,y\right)*o\left(x,y\right)\right]+\nabla_{o}p\left(o,h\right)=0$$
(12)$$o^{c}\left(x,y\right)*\left[g\left(x,y\right)-h\left(x,y\right)*o\left(x,y\right)\right]+\nabla_{h}p\left(o,h\right)=0$$

Multiply both sides of (Equation 11) and (Equation 12) by a positive actual number λ, which is a parameter used to adjust the convergence rate of the algorithm. When it is large, the algorithm converges quickly. This paper adopts the sigmoid function to promote the optimization performance
(13)$$2Sigmoid(\lambda_{1}h^{c}\left(x,y\right)*\left[g\left(x,y\right)-h\left(x,y\right)*o\left(x,y\right)\right]+\nabla_{o}p\left(o,h\right))=1$$
(14)$$2Sigmoid(\lambda_{2}o^{c}\left(x,y\right)*\left[g\left(x,y\right)-h\left(x,y\right)*o\left(x,y\right)\right]+\nabla_{h}p\left(o,h\right))=1$$

Multiply both sides of (Equation 13) and (Equation 14) by the estimates of o(x,y) and h(x,y), respectively, to arrive at the final iterative formulae for image restoration.   
(15)$$\begin{array}{cc} o_{k+1}\left(x,y\right) & =2o_{k}\left(x,y\right)Sigmoid\left(\frac{\lambda_{1}\partial J\left(o_{k},h_{k}\right)}{\partial o_{k}\left(x,y\right)}\right)\\ & =2o_{k}\left(x,y\right)Sigmoid\left(\lambda_{1}h_{k}^{c}\left(x,y\right)*\left[g\left(x,y\right)-h_{k}\left(x,y\right)*o_{k}\left(x,y\right)\right]+\nabla_{o}p\left(o,p\right)\right),\lambda_{1}>0 \end{array}$$
(16)$$\begin{array}{cc} h_{k+1}\left(x,y\right) & =2h_{k}\left(x,y\right)Sigmoid\left(\frac{\lambda_{2}\partial J\left(o_{k},h_{k}\right)}{\partial h_{k}\left(x,y\right)}\right)\\ & =2h_{k}\left(x,y\right)Sigmoid\left(\lambda_{2}o_{k}^{c}\left(x,y\right)*\left[g\left(x,y\right)-h_{k}\left(x,y\right)*o_{k}\left(x,y\right)\right]+\nabla_{h}p\left(o,p\right)\right),\lambda_{2}>0 \end{array}$$

For (Equation 15) and (Equation 16), this paper initializes o(x,y) and h(x,y) to the matrices of all ones due to their insufficiency. In this paper, in order to make the result converge and protect the edge information of the image while removing the noise, Equations (Equation 15) and (Equation 16) can be rewritten as
(17)$$\begin{array}{cc} \; & o_{k+1}\left(x,y\right)=2o_{k}\left(x,y\right)Sigmoid(\lambda_{1}h_{k}^{c}\left(x,y\right)*[g\left(x,y\right)-h_{k}\left(x,y\right)*o_{k}\left(x,y\right)*\\ & \left(1+\mu h_{SobelV}\left(x,y\right)*h_{SobelH}\left(x,y\right)\right)*h_{GaussianLP}\left(x,y\right)]+\nabla_{o}p\left(o,p\right)) \end{array}$$
(18)$$\begin{array}{cc} \; & h_{k+1}\left(x,y\right)=2h_{k}\left(x,y\right)Sigmoid(\lambda_{2}o_{k}^{c}\left(x,y\right)*[g\left(x,y\right)-h_{k}\left(x,y\right)*o_{k}\left(x,y\right)*\\ \; & \left(1+\mu h_{SobelV}\left(x,y\right)*h_{SobelH}\left(x,y\right)\right)*h_{GaussianLP}\left(x,y\right)]+\nabla_{h}p\left(o,p\right)) \end{array}$$

The $h_{GaussianLP}\left(x,y\right)$ represents the Gaussian low-pass filter; $h_{SobelV}\left(x,y\right)$ is the Sobel vertical edge detector impulse response function; $h_{SobelH}\left(x,y\right)$ is the Sobel horizontal edge detector impulse response function. μ∈[0.15,0.35] is the edge protection factor. This paper chose a more considerable value when there are many details in the image; otherwise, it chose a smaller one. λ∈[600,1200] is the coefficient that controls the convergence rate. When λ takes a considerable value, the convergence speed is fast.

The Sobel vertical edge detector impulse response function, $h_{SobelV}\left(x,y\right)$, defined as
(19)$$h_{SobelV}\left(x,y\right)=\left\{\begin{array}{cc} sgn(x) & if\;|y|=1\;and\;|x|=1\\ 2sgn(x) & if\;|y|=0\;and\;|x|=1,\\ 0 & otherwise \end{array}\right.$$
and the Sobel horizontal edge detector impulse response function, $h_{SobelH}\left(x,y\right)$, defined as
(20)$$h_{SobelH}\left(x,y\right)=\left\{\begin{array}{cc} sgn(y) & if\;|x|=1\;and\;|y|=1\\ 2sgn(y) & if\;|x|=0\;and\;|y|=1,\\ 0 & otherwise \end{array}\right.$$
where sgn(·) denotes the sign function, i.e.,
(21)$$sgn\left(x\right)=\left\{\begin{array}{cc} 1 & for\;x>1\\ 0 & for\;x=1,\\ -1 & for\;x<1 \end{array}\right.$$

The Gaussian low-pass filter, $h_{GaussianLP}\left(x,y\right)$, defined as
(22)$$h_{GaussianLP}\left(x,y\right)=\frac{1}{2\pi \sigma^{2}}e^{-\frac{x^{2}+y^{2}}{2\sigma^{2}}}$$

The support size of Gaussian low-pass filter is fixed as 5 × 5, and the standard deviation σ is set to 0.5–2.0. The range σ is a user parameter, which is related to the noise level of the input image. When images contains much noise, a large σ is chosen. For simplicity, drop “(x,y)” in (Equation 17) and (Equation 18),
(23)$$\begin{array}{c} o_{k+1}=2o_{k}Sigmoid\left(\lambda_{1}h_{k}^{c}*\left[g-h_{k}*o_{k}*\left(1+\mu h_{SobelV}*h_{SobelH}\right)*h_{GaussianLP}\right]+\nabla_{o}p\left(o,h\right)\right) \end{array}$$
(24)$$\begin{array}{c} h_{k+1}=2h_{k}Sigmoid\left(\lambda_{2}o_{k}^{c}*\left[g-h_{k}*o_{k}*\left(1+\mu h_{SobelV}*h_{SobelH}\right)*h_{GaussianLP}\right]+\nabla_{h}p\left(o,h\right)\right) \end{array}$$

Iterating Equations (Equation 23) and (Equation 24) to alternately estimate the $o_{k}\left(x,y\right)$ and $h_{k}\left(x,y\right)$, achieve the maximum of Equation (Equation 3) and obtain the best original image estimation. The main steps of the proposed BDA-SF are shown in the Algorithms 1 and 2.
**Algorithm 1** Estimate latent image**Input:** Blurred image *g*, kernel estimation $h_{0}$, regularization weight α, γ, parameter λ, iterations *J*, *K*;1: $o_{k}\leftarrow g$, $h_{k}\leftarrow h_{0}$.2: **while**
iter<K
**do**3:     **if**
iter<J
**then**4:         **for**
iter=0:J−1
**do**5:            Compute $o_{k+1}$ via (Equation 17) using $h_{k}$, $o_{k}$;6:            Compute $h_{k+1}$ via (Equation 18) using $h_{k}$, $o_{k}$;7:         **end for**8:     **else**
J<iter<K9:         **for**
iter=0:K−1
**do**10:            Compute $o_{k+1}$ via (Equation 17) using $h_{k}$, $o_{k}$;11:            Compute $h_{k+1}$ via (Equation 18) using $h_{k}$, $o_{k}$;12:         **end for**13:     **end if**14: **end while****Output:** Intermediate latent image *o*. Blur kernel *h*.

**Algorithm 2** Estimate Blur kernel**Input:** Blurred image *g*, maximum iterations *K*.1: **while**
iter<K
**do**2:     Update latent image *o* via Algorithm 1;3:     Update blur kernel *h* via (Equation 18);4: **end while****Output:** Intermediate latent image *o*. Blur kernel *h*.

### 3.2. Sigmoid Function

It is the sigmoid function of the proposed BDA-SF that provides the critical difference from the MIA which uses the exponential function, and significantly improves the blind deconvolution performance. For comparison, these two functions are plotted here and shown in Figure 1. Figure 1a shows the plots of exponential functions with different coefficients, while Figure 1b shows a cluster of sigmoid functions. Figure 1 shows that the exponential function is asymmetric. For the negative variable, it changes slowly, while for the positive variable, it changes steeply. That is to say, the exponential function may fail to update the estimator and tend towards zero when $h^{c}\left(x,y\right)*\left[g\left(x,y\right)-h\left(x,y\right)*o\left(x,y\right)\right]$ is much less than zero. While the $h^{c}\left(x,y\right)*\left[g\left(x,y\right)-h\left(x,y\right)*o\left(x,y\right)\right]$ is much bigger than zero, the estimator may overly update, thus incurring an enormous negative value in the next iteration. We suppose this exponential function phenomenon is why MIA cannot be applied to severely degraded image blind deconvolution. Conversely, the sigmoid function is symmetric. It is free from the problems of the exponential function mentioned above. Further, its saturation property helps it deal with the immense value of the variable. Therefore, benefiting from these properties of the sigmoid function, this blind deconvolution algorithm performs well, especially with severely degraded images.

**Image Sequence**: The image target scene changes little when imaging with a short exposure (the imaging system tends to have a high frame rate). Therefore, this paper considers that adjacent frames are similar to the same target image and different point spread functions (PSF). Figure 2 describes the image degradation process.

It is reasonable to assume that adjacent frames do not change significantly in a short period [51]. For image degradation by atmosphere turbulence, the displacement of the target image mainly comes from the degradation caused by turbulence rather than the target itself changing. Figure 3 simulates the image degradation caused by atmospheric turbulence. These blur kernels are generated by the random phase screen [54]. The parameters of the simulated atmosphere turbulence were chosen to create images similar to images recorded by telescope (D = 1.50 m) through a turbulence of $r_{0}$ = 0.045–0.055. The blur nearly occupies 25 × 25 pixels in the 128 × 128 pixels image pane. Similarly, the assumption that the target image will not change in the short-term is also applicable to other situations where the imaging frame rate is high, such as removing motion blur.

The specific process of the algorithm is as follows: The work divides the reconstruction of the iterative algorithm into two stages. The first stage is to restore the original sequence. The second stage is the restoration of the remaining sequences. As shown in Figure 4, this paper assumes that the short exposure image sequence does not change much in a short period. This paper treats the first five frames of the input image sequence as a sub-sequence and iterated the frames several times to restore them. The appropriate number of frames needs to be selected according to the target scene. When dealing with a single image, set J = 0. The method is suitable for short exposure images with little change in the target. BDA-SF obtains an average result of the sub-sequence. Using the average value can prevent unknown noise interference. BDA-SF uses this result as the initial estimate for subsequent frames. Next, BDA-SF uses the result of the previous frame as the initial estimate for the next frame. In this way, BDA-SF can obtain good results with fewer iterations.

## 4. Experimental Results and Analysis

First, This paper provides a practical application of the algorithm and analyzes the convergence of the algorithm. Second, this paper compares the algorithm with traditional algorithms. Third, this paper compares the algorithm with state-of-the-art methods.

### 4.1. Performance Evaluation

To evaluate the result of restored images. This paper uses the peak signal-to-noise ratio (PSNR) [55] and structural similarity (SSIM) [56] to evaluate the effect of image restoration.

PSNR is the peak value of the signal to noise in the images. The equation is shown below
(25)$$PSNR=10log10\frac{MAX_{o}^{2}}{\left|\right|\hat{o}-o{\left|\right|}_{2}^{2}}$$
where *o* is the latent image. $\hat{o}$ is the restored image. $MAX_{o}$ is the maximum value of the image *o*.

SSIM is used to evaluate the degree of similarity of geometric structure information of the restored image and the latent image. The equation is as below
(26)$$SSIM=\frac{\left(2\mu_{o}\mu_{\hat{o}}+C_{1}\right)\left(2\sigma_{o\hat{o}}+C_{2}\right)}{\left(\mu_{o}^{2}+\mu_{\hat{o}}^{2}+C_{1}\right)\left(\sigma_{o}^{2}+\sigma_{\hat{o}}^{2}+C_{2}\right)}$$
where $\mu_{o}$, $\mu_{\hat{o}}$ denote the means of *o*, $\hat{o}$, respectively. $\sigma_{o}$, $\sigma_{\hat{o}}$ denote variances of *o*, $\hat{o}$, respectively. $\sigma_{o\hat{o}}$ is the image covariance.

### 4.2. Convergence Property

Figure 5 shows the frames from one video sequence of a flying plane. This paper converts the frames in the video sequence to 256 grayscale for convenience and sheared the images to 256 × 256 pixels. Figure 5a is the initial frame. This paper sets the parameters as μ=0.25, $\lambda_{1}=800$, $\lambda_{2}=1000$, α=0.04, γ=2. Through 200 iterations of the algorithm, we obtain Figure 5d. At this time, the picture quality was not improved significantly. The goal of this step is to obtain an initial estimate. Figure 5b is the 20th frame of the sequence. BDA-SF iterates Figure 5e 40 times to get Figure 5b. At this point, we can vaguely see numbers on the fuselage of the plane. The picture quality has improved to some extent. Figure 5c is the 40th frame of the video sequence. Restoring Figure 5c obtains Figure 5f. The picture quality was greatly improved. We can see the number “126” on the fuselage. With the deepening of iteration, BDA-SF can restore the image sequence efficiently. The most time-consuming part of the algorithm is the Fourier transform. The complexity of the algorithm is O(nlogn). The simulations are carried out on Windows 10 with an Intel Core i5-7200U CPU at 2.7 GHz with 12 GB RAM. The algorithm takes about 0.04 s per iteration to process the 256 × 256 image. Using the previous frame as the initial estimate can save many iterative steps and improve the algorithm’s efficiency.

Figure 6 shows the iterative curve. The horizontal axis represents the number of iteration, and the vertical axis is the residual. The black line is a direct iterative algorithm. The curve marked by a red star represents the first stage of the algorithm. The first stage of restoration did not arrive at the optimal point, but this does not matter; all we need is an initial estimate. The next step needs a few iterations, and the green line represents the second step of the algorithm. BDA-SF can achieve convergence with only a few iterations—no more than 20. Although the first step requires lots of iterations, it reduces the number of iterations needed for later work.

To better show the convergence of the algorithm. This paper randomly selects four-pixel points in Figure 5f and investigates the change in their pixel values with the number of iterations. At the same time, this paper also obtains the residual curve of the image. Figure 7 shows the variation in pixel values and residual. Figure 7a shows the pixel values with the number of iterations; Figure 7b is the residual curve of the image.

### 4.3. Compared with Traditional Methods

It can be seen from Figure 8 that the proposed algorithm can protect the edge details of the image while removing the blur. BDA-SF restores and extends the spectrum, and the image quality is improved. The algorithm achieves high-resolution restoration.

**Another example:** This paper restores the tower from an actual video. Figure 9a–c are the blurred images and Figure 9d–f are the restored images. Figure 9 shows that the texture information obtained is abundant. Even the lines on the top of the tower are clear.

### 4.4. Compared with State-of-the-Art Methods

To better evaluate the algorithm. This paper selects severely degraded images from the public dataset [18], which contains four images and eight kernels. Figure 10 shows the comparison between the proposed algorithm and other iterative algorithms based on MAP estimates. Algorithms involved in the comparison are Krishnan et al. [19], Xu et al. [20], Pan et al. [22], Yan et al. [27], Jin et al. [31], Bai et al. [32]. This paper uses evaluation indexes PSNR and SSIM to evaluate the image quality. Table 1 provides a quantitative evaluation of Figure 10. Table 1 shows that the image restored by the method has the highest PSNR and SSIM. [19] has the best visual effect, but it is too sharp compared to the original image, resulting in poor evaluation. This paper also shows the error ratios for various algorithms in Figure 11. Figure 11 shows that BDA-SF can achieve 100% success at an error ratio of 2.

Figure 12 is from the dataset by Kohler et al. [57] containing four images and twelve kernels. This paper chooses a severely degraded image from the dataset. The compared algorithms include Xu et al. [15], Krishnan et al. [19], Whyte et al. [30], Xu et al. [20], Pan et al. [23], Pan et al. [22], Yan et al. [27], Jin et al. [31], Bai et al. [32]. Table 2 provides a quantitative evaluation of Figure 12. Figure 13 investigates the effectiveness of the sigmoid function. The results demonstrate that the sigmoid function gives rise to significant SSIM (Figure 13b) and PSNR (Figure 13a) improvement. Figure 14 presents the PSNR results of the compared algorithms. Figure 14 shows that BDA-SF can achieve a state-of-the-art performance. It can be inferred from Figure 12 and Figure 14 that BDA-SF can achieve comparable visual results compared with the state-of-the-art methods [22,27]. Reference [27] is slightly superior to BDA-SF in PSNR and SSIM. Figure 12 is a dark scene with lights, reference [27] used the dark channel and the bright channel at the same time, so [27] achieved the best results. However, reference [27] has poor robustness and may perform poorly on other images, such as Figure 15e.

This paper evaluates the method on natural, face, text, and low-illumination images. This paper also reports results on images with non-uniform blur. This paper provides typical results for each class. Finally, this paper also compares the running time of different algorithms.

**Natural image:** The natural images are from the dataset [57]. Figure 15 presents a visual comparison. The algorithm achieves competitive results against the method [23]. Furthermore, The method has a better visual result on textures for the local details than other state-of-the-art methods.

**Face image:** Face image deblurring is a challenge for algorithms designed for natural images. The lack of textures and edges in face images makes kernel estimation challenging. It can be inferred from Figure 16 that the method can achieve comparable visual results to other methods [23,27].

**Text image:**Figure 17 illustrated the results of the state-of-the-art methods on a text image. The algorithm can achieve a superior performance compared with existing methods. Visually, BDA-SF shows better texture features compared with the method [23]. While methods [19,20,31] produce heavy ringing artifacts, BDA-SF achieves more explicit images.

**Low-illumination image:** It is particularly challenging for most deblurring methods to deal with the low-illumination images because low-illumination images often have saturated pixels that interfere with kernel estimation. Figure 18 shows the results of the state-of-the-art methods on a low-illumination image. As a result, the method achieves a comparable result with the method [21], designed explicitly for low-illumination images.

**Non-uniform deblurring:** This paper applies the method to non-uniform blur. Figure 19 presents the results on images degraded by spatially variant blur. It can be inferred from Figure 19 that BDA-SF can give comparable visual results to the state-of-the-art non-uniform deblurring method [20,30]. Figure 20 shows the results and their corresponding intermediate results. With the sigmoid function, the results contain more sharp edges and texture features.

**Computation complexity:** This paper compares the computation complexity of BDA-SF with existing state-of-the-art methods [19,20,23,27,31]. The simulations are carried out on Windows 10 with an Intel Core i5-7200U CPU at 2.7 GHz with 12 GB RAM. The natural image size is 280 × 325; face image size is 284 × 365; text image size is 1097 × 1094; low-illumination image size is 800 × 533. The runtime of the non-blind deblurring step includes the total time. Among the methods, it can be seen from Table 3 that the method developed by Krishnan et al. [19] is the fastest. However, its results are inferior to BDA-SF, as illustrated above. BDA-SF is slower than the method [23]. BDA-SF is twice as fast as the method [31].

### 4.5. Effectiveness of BDA-SF

BDA-SF is based on sigmoid function, which constructs novel blind deconvolution estimators for both original image and degradation process. Figure 5 and Figure 9 are applications of BDA-SF. Figure 8 demonstrates that BDA-SF can protect the edge details concerning the Sobel filter (μ=0.25).

To better evaluate the effectiveness of the sigmoid function, this paper selects severely degraded images from public datasets [18,57]. Methods involved in the comparison are Xu et al. [15], Krishnan et al. [19], Whyte et al. [30], Xu et al. [20], Pan et al. [23], Pan et al. [22], Yan et al. [27], Jin et al. [31], Bai et al. [32]. Figure 11 and Figure 12 show the results. This paper uses evaluation indexes PSNR and SSIM to evaluate the image quality. Table 1 and Table 2 demonstrate that BDA-SF using sigmoid function can achieve a state-of-the-art performance on severely degraded images.

This paper evaluates the method on natural, face, text, and low-illumination images. To better evaluate the effectiveness of the sigmoid function, ablation experiments were performed. As is shown in Figure 15, Figure 16, Figure 17 and Figure 18, the images recovered using sigmoid function are more visually pleasing. Figure 19 shows that BDA-SF using sigmoid function generates intermediate results with more sharp edges. All the results demonstrate the effectiveness of the sigmoid function.

### 4.6. Limitation

This paper establishes the likelihood function assuming that the noise obeys Gaussian distribution. When the image has significant non-Gaussian noise, the algorithm cannot achieve satisfactory results. Figure 21 shows an example of the method dealing with images degraded by salt and pepper noise. As shown in Figure 21, the method will not work well when dealing with images degraded by non-Gaussian noise. Another drawback of the method is that the running speed is not fast enough. Table 3 demonstrates that the algorithm is slower than [19,27]. In the future, we will consider the effects of various noises (such as salt and pepper noise). We will also consider extending the algorithm to video deblurring.

## 5. Conclusions

This paper proposes a new iterative algorithm based on the sigmoid function for image restoration. The algorithm can naturally maintain the non-negative constraint of the solution during the restoration process. The algorithm can effectively enhance the high frequency spectrum and achieve high-resolution restoration, even when images are severely degraded. Since all operations in the algorithm are multiplication operations, the method can avoid the instability of numerical calculations. The approach has added a low pass filter and edge-preserving process to the iteration formulae, to protect the image’s edges while removing noise sufficiently. For the image sequence, the method uses inter-frame information, from which satisfactory results can be obtained with fewer iterations. Extensive experiments demonstrate that the method achieves a state-of-the-art performance for both natural images and images acquired under specific scenarios. It is expected that the success of deploying the sigmoid function in construction of the blind deblurring algorithm will motivate further research in the field of image restoration.

## Figures and Tables

**Figure 1 sensors-21-03484-f001:**
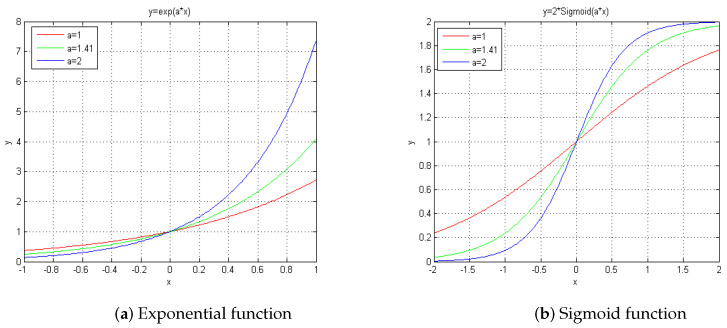
Comparison between exponential function and Sigmoid function.

**Figure 2 sensors-21-03484-f002:**
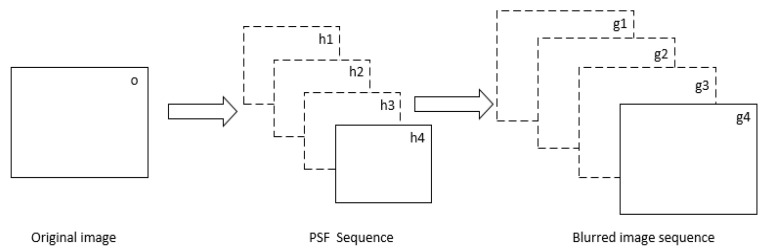
Image degradation under short exposure conditions.

**Figure 3 sensors-21-03484-f003:**
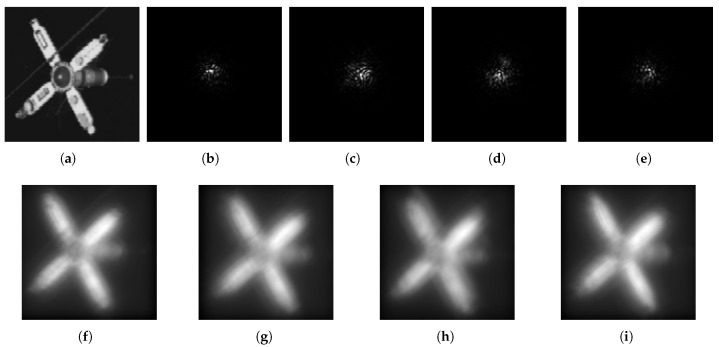
The image is degraded by the point spread function. (**a**) is the original image. Resolution is 128 × 128. Point Spread Functions are shown in (**b**–**e**). Degraded images corresponding to the (**b**–**e**) are shown in (**f**–**i**).

**Figure 4 sensors-21-03484-f004:**
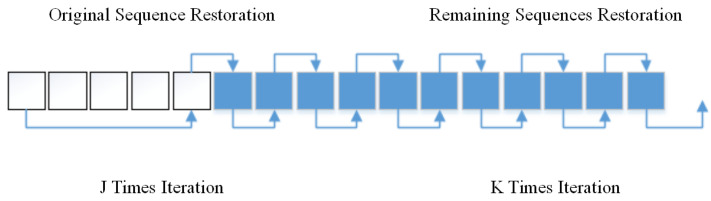
Two-step iterative algorithm. The iteration time J is greater than K.

**Figure 5 sensors-21-03484-f005:**
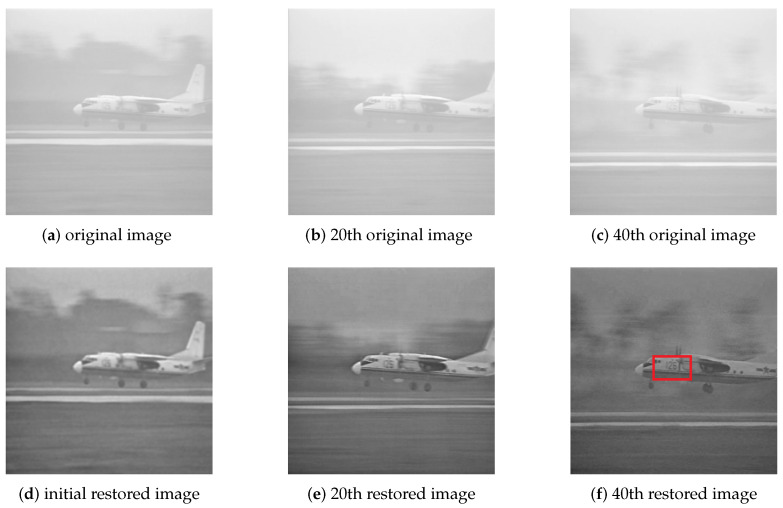
Frames in a video sequence.

**Figure 6 sensors-21-03484-f006:**
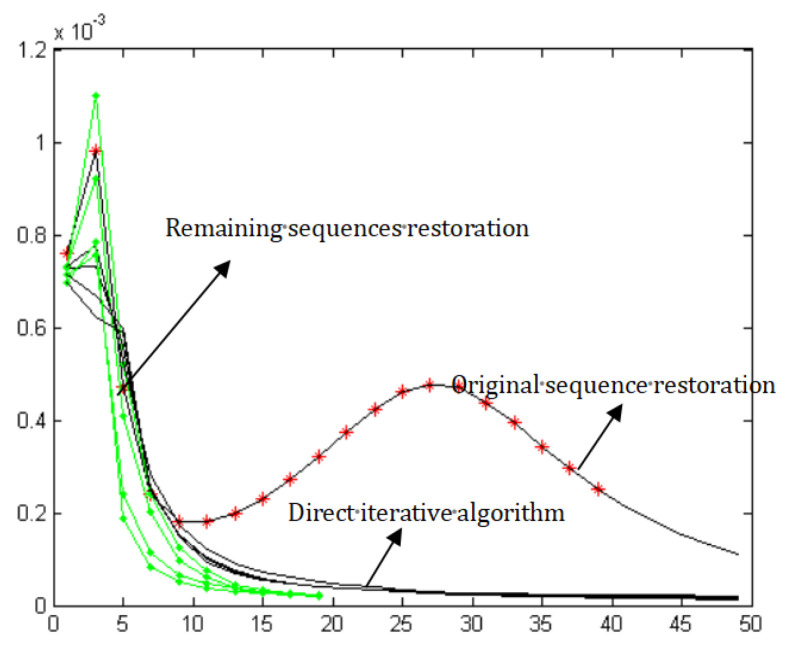
Iterations curve.

**Figure 7 sensors-21-03484-f007:**
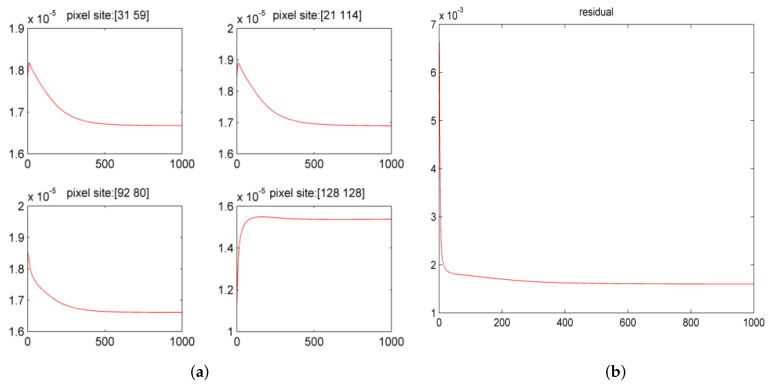
Pixel value curve and residual curve. The horizontal direction represents the number of iterations. The vertical direction represents the pixel value.

**Figure 8 sensors-21-03484-f008:**
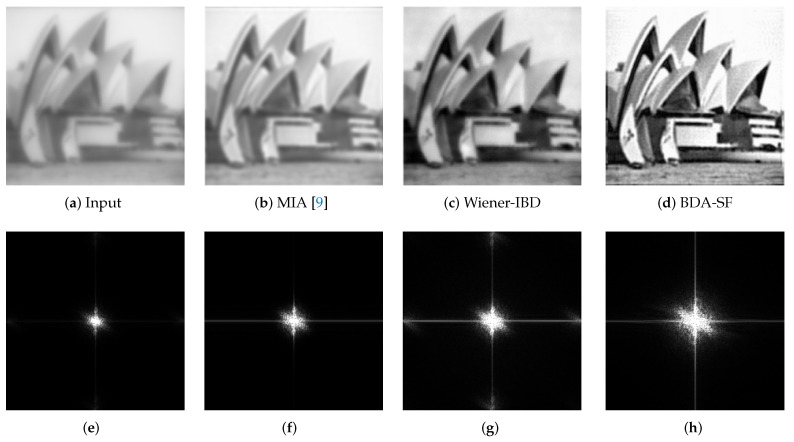
The proposed algorithm is compared with other general algorithms. (**a**) an actual blurred image. (**b**) Multiplicative iterative algorithm. (**c**) Wiener-IBD. (**d**) the proposed BDA-SF. (**e**–**h**) are the corresponding spectra.

**Figure 9 sensors-21-03484-f009:**
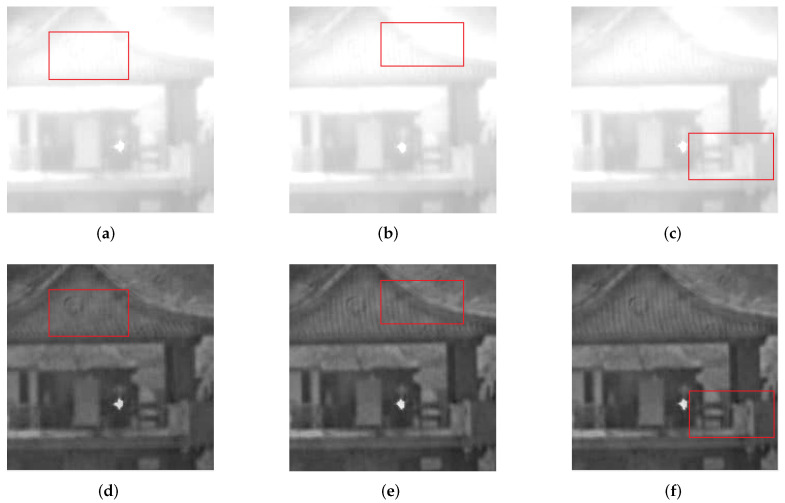
The tower in the actual video.

**Figure 10 sensors-21-03484-f010:**
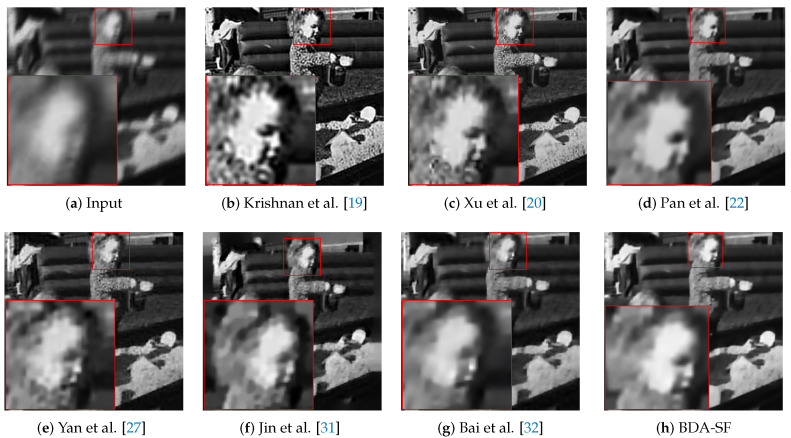
Deblurred results from the dataset [18]. The PSNR and SSIM values are shown in Table 1. BDA-SF has the highest PSNR and SSIM. (Best viewed on high-resolution display with zoom-in).

**Figure 11 sensors-21-03484-f011:**
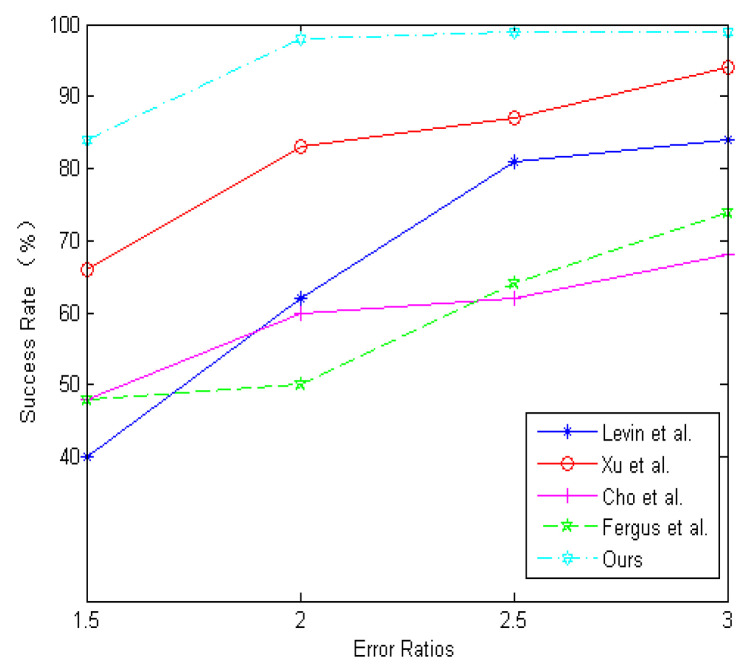
Comparisons in terms of cumulative error ratio.

**Figure 12 sensors-21-03484-f012:**
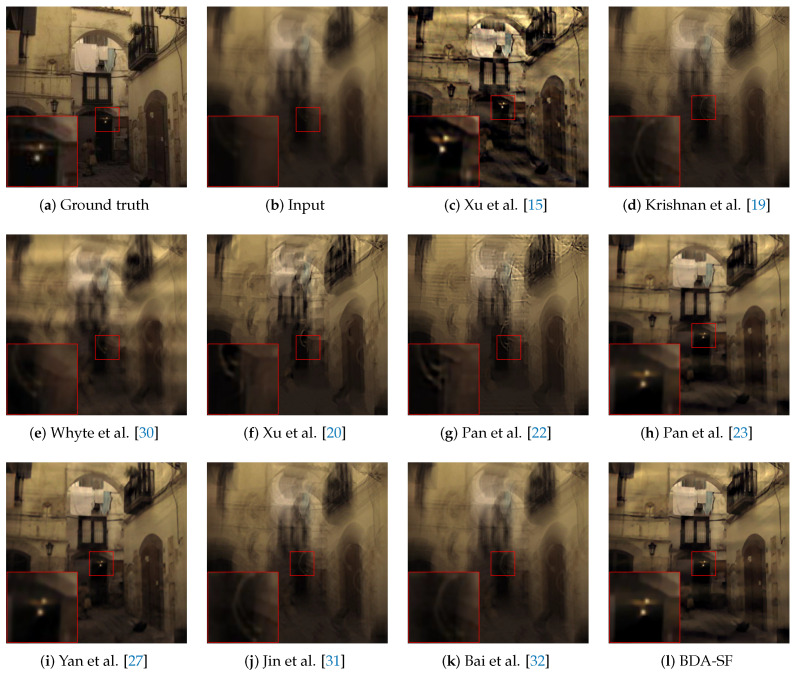
Deblurred results from the dataset [57]. The PSNR and SSIM values are shown in Table 2. BDA-SF has the second highest PSNR and SSIM. The deblurred image estimated by BDA-SF is visually more pleasing. (Best viewed on high-resolution display with zoom-in).

**Figure 13 sensors-21-03484-f013:**
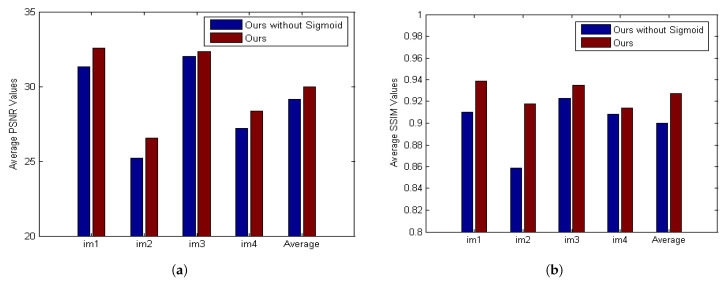
Quantitative evaluation results of BDA-SF.

**Figure 14 sensors-21-03484-f014:**
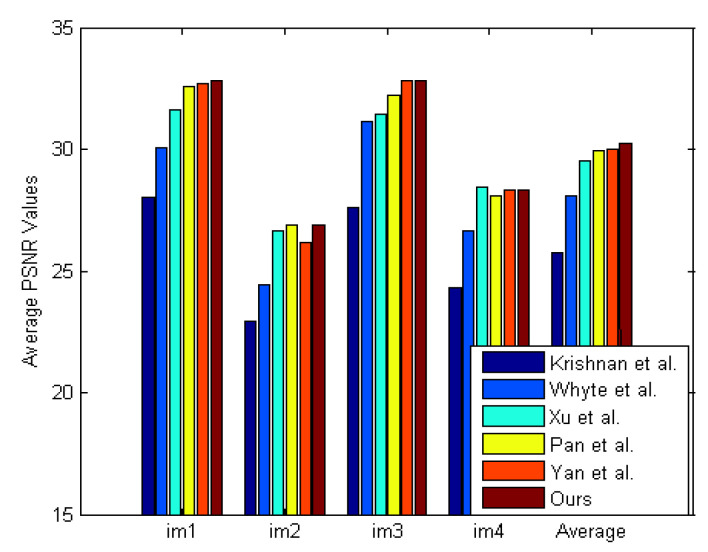
Quantitative evaluation results on the dataset [57].

**Figure 15 sensors-21-03484-f015:**
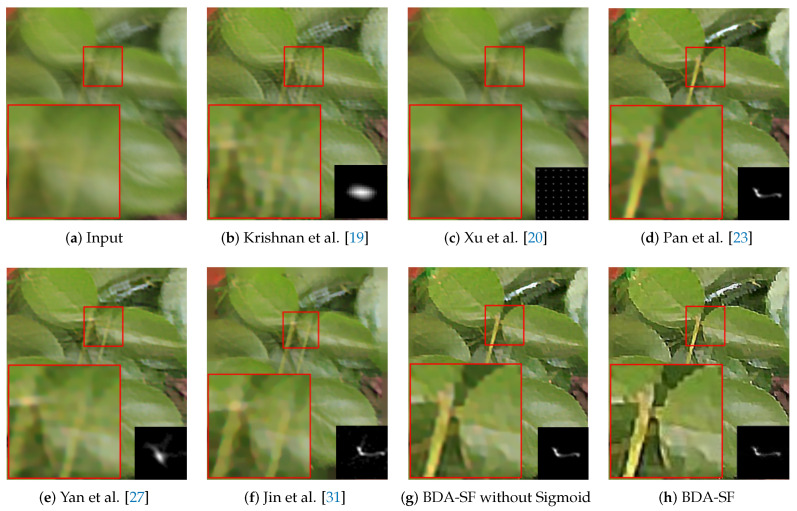
Visual comparison on a real natural image. BDA-SF achieves finer edges and details, as is shown in red boxes. (Best viewed on high-resolution display with zoom-in).

**Figure 16 sensors-21-03484-f016:**
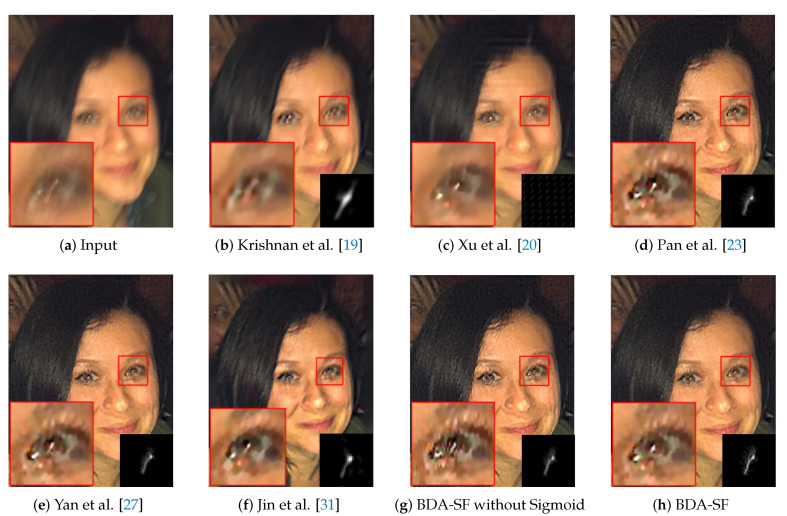
Visual comparison on a face image. BDA-SF achieves comparable visual results with method [23,27,31]. (Best viewed on high-resolution display with zoom-in).

**Figure 17 sensors-21-03484-f017:**
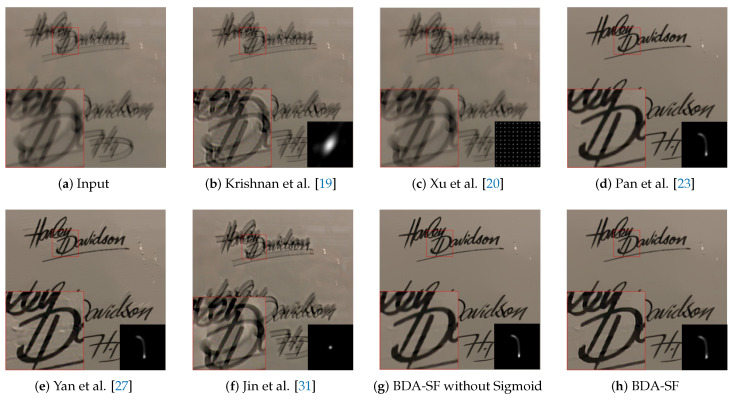
Visual comparison on a text image. BDA-SF achieves comparable visual results with method [23,27]. (Best viewed on high-resolution display with zoom-in).

**Figure 18 sensors-21-03484-f018:**
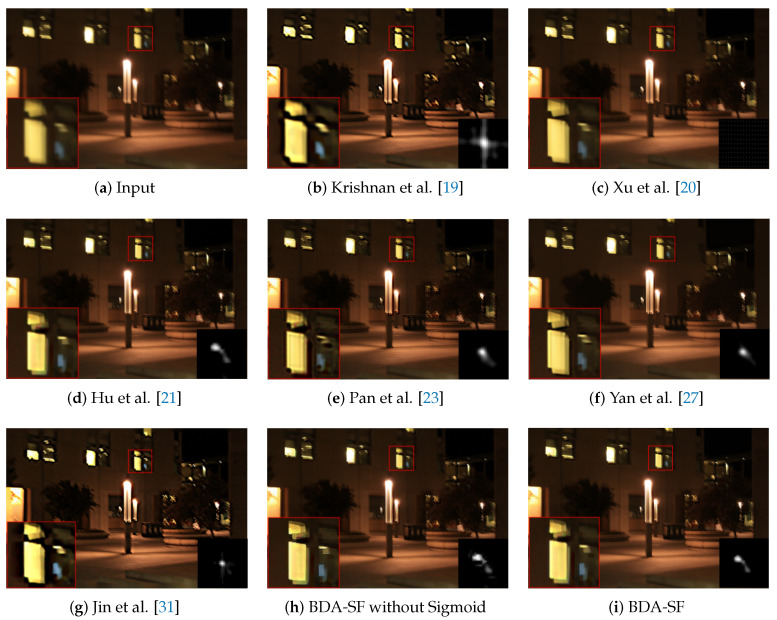
Visual comparison on a low-illumination image. BDA-SF achieves comparable visual results with method [21] which is specifically designed for low-illumination images. (Best viewed on high-resolution display with zoom-in).

**Figure 19 sensors-21-03484-f019:**
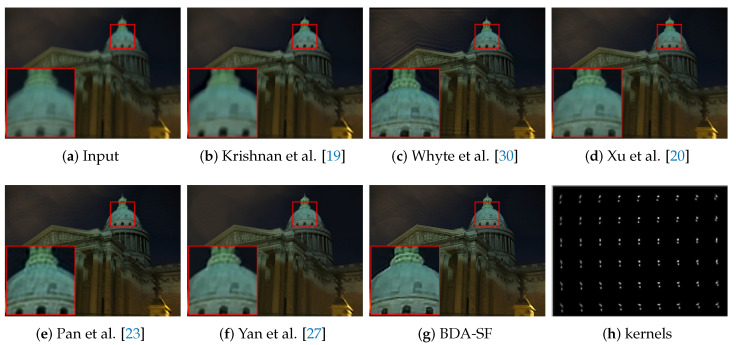
Visual comparison on images with non-uniform blur. Kernels are resized for visualization. BDA-SF is visually comparable to methods [20]. Method [23] contains ringing artifacts and residual blurs.

**Figure 20 sensors-21-03484-f020:**
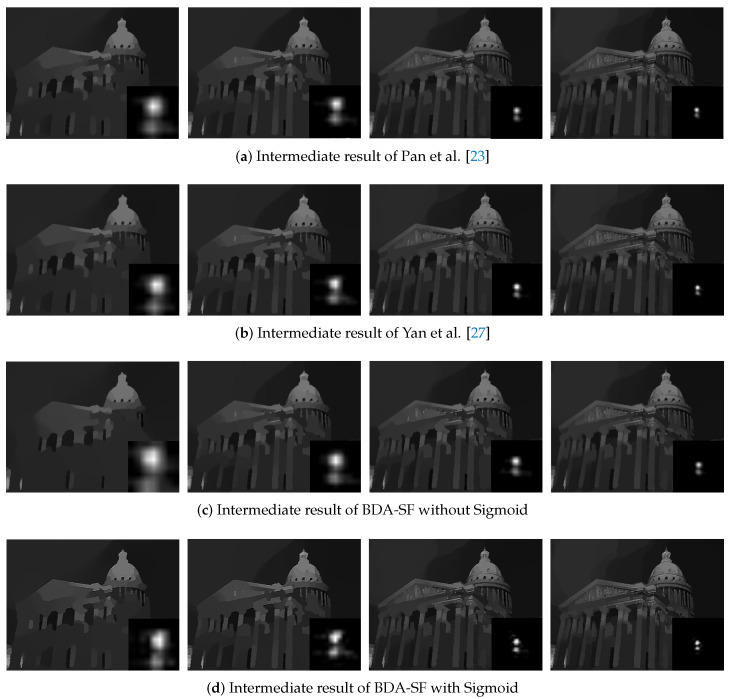
Deblurred results and its corresponding intermediate results over iterations. With the Sigmoid function, the proposed BDA-SF achieves intermediate results containing more sharp edges. The use of the Sigmoid function makes the results contain sharper edges and texture features.

**Figure 21 sensors-21-03484-f021:**
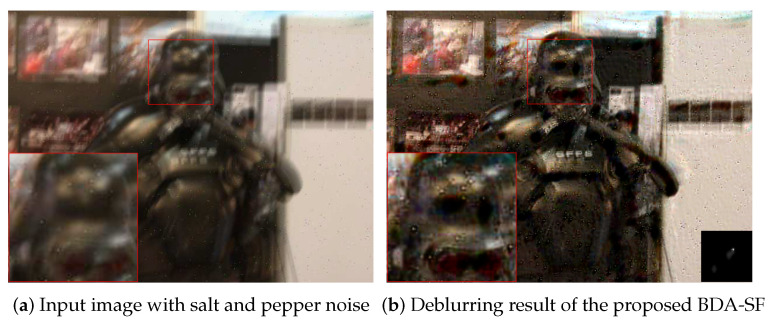
Limitation of the proposed model.

**Table 1 sensors-21-03484-t001:** Quantitative evaluations on the image from Figure 10.

Methods	PSNR	SSIM
Krishnan et al. [19]	21.2398	0.7588
Xu et al. [20]	20.8402	0.6921
Pan et al. [22]	19.2688	0.6089
Yan et al. [27]	24.2150	0.7683
Jin et al. [31]	23.8377	0.7542
Bai et al. [32]	26.4120	0.8174
BDA-SF	27.2434	0.8859

**Table 2 sensors-21-03484-t002:** Quantitative evaluations on the image from Figure 12.

Methods	PSNR	SSIM
Xu et al. [15]	19.0964	0.6987
Krishnan et al. [19]	21.9974	0.8330
Whyte et al. [30]	20.6246	0.8254
Xu et al. [20]	21.8491	0.8373
Pan et al. [22]	21.7723	0.8250
Pan et al. [23]	23.9403	0.8047
Yan et al. [27]	25.5430	0.8507
Jin et al. [31]	22.0974	0.8376
Bai et al. [32]	22.0311	0.8401
BDA-SF	25.0137	0.8413

**Table 3 sensors-21-03484-t003:** Runtime (in seconds) of different methods. The code is implemented in MATLAB.

Methods	280 × 325	284 × 365	1097 × 1094	800 × 533
Krishnan et al. [19]	20.41	23.71	156.40	74.37
Xu et al. [20]	226.51	468.56	4033.79	1655.50
Pan et al. [23]	319.34	295.60	4078.68	1201.16
Yan et al. [27]	47.99	46.50	1077.90	294.89
Jin et al. [31]	561.12	620.04	14187.65	2814.74
BDA-SF	255.45	249.95	3115.04	1075.10

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
