# Peer review of "Blind Deblurring Based on Sigmoid Function"

_sensors, 2021, doi:10.3390/s21103484_

Round 1
Reviewer 1 Report
See comments on math in the attached file.
- Abstract: “Blind image deblurring, also known as blind image deconvolution, is a long-standing challenge in the field of image processing and low-level vision. To restore a clear version of the severely degraded image, this paper proposes a blind deblurring algorithm based on the sigmoid function, which constructs novel blind deblurring estimators for both the original image and degradation process by exploring the excellent property of sigmoid function and considering image derivative constraints. Owing to these symmetric and non-linear estimators of low computation complexity, high quality images can be obtained by the algorithm. The algorithm is also extended to image sequences. The sigmoid function enables the proposed algorithm to achieve state-of-the-art performance in various scenarios, including natural, text, face, and low-illumination images. Besides, the method can be extended naturally to non-uniform deblurring. Quantitative and qualitative experimental evaluations indicate that the algorithm can remove the blur effect and improve the image quality of actual and simulated images. Finally, the use of sigmoid function provides a new approach to algorithm performance optimization in the field of image restoration.”
- “…to restore the clear picture from its blurred counterparts with unknown blur parameters…”
- “In image restoration literature, image degradation is commonly modeled as follows [1]:
g(x, y) = o(x, y) ∗ h(x, y) + n(x, y) (1)
where “*” is the convolution operator, o(x, y) and g(x, y) stand for the given clear image and its degraded counterpart, respectively, h(x, y) denotes the point spread function (PSF) representing degradation induced in the spatial domain, and n(x, y) represents the additive noise.”
While the authors cited [1] by A.K. Jain in the manuscript, it seemed the formulation shown by (1) was more in line with that given in the textbook Digital Image Processing, 4th ed. by R.C. Gonzalez and R.E. Woods, 2018.
- “The approach can obtain a good result quickly for image restoration algorithms in the frequency domain [3].”
- “After completing data processing in the frequency domain, it converts the results to the spatial domain.”
- “The most critical probabilistic models are maximum likelihood estimation (MLE) and maximum a posterior estimation (MAP) [6], by which the image restoration problem is converted to probability estimation through Bayesian inference.”
- “A multiplicative iterative approach (MIA) [8], [9] was proposed based on a probabilistic model. MIA [8], [9] naturally preserves the non-negative constraint on the iterative solutions when the initial estimates are non-negative, producing a restored image of high quality.”
- “…The estimation of adjacent frames of short exposure sequence is applied to approximate the current frame and to obtain a better target image estimation. …”
- “The contributions are as follows. First, this paper proposes an image restoration model based on the Sigmoid function. Intuitively, the latest iteration model can ensure that the image is non-negative in the iteration process. As a result, it does not need any other constraints to make the pixel value non-negative. Second, the approach can effectively restore severely degraded images using the sigmoid function and the information between sequences. Compared with the classical and the state-of-the-art methods, experiments show that the new method has better competitive performance for severely degraded images. Third, to better evaluate the algorithm’s performance, this paper presents more experimental results of blind deblurring. The results demonstrate that the new algorithm can achieve the performance of state-of-the-art methods.”
- “…In the first step, the blur kernel is estimated by using the image obtained….”
Change “blurred kernel” to “blur kernel” throughout the manuscript.
Define “blur kernel” and cite relevant reference(s).
- “…Having reviewed image restoration progress of the last decade in this section, the remaining contents of this article are organized as follows. In Section 3, …”
- “…Based on the idea of the multiplicative iterative algorithm (MIA), which is efficient but limited to weak degradation, a novel blind deconvolution algorithm is devised employing the sigmoid function, i.e., the BDA-SF, for restoration of seriously degraded images to overcome MIA’s limitation. The algorithm has good convergence with simple parameter selection. …”
- “It has been shown that the performance of the least-squares algorithm is almost insensitive to whether noise is Poissonian or Gaussian [48], and that for Poissonian noise, no strong difference exists between the results of the ISRA and those of the RLA, while for Gaussian noise, the ISRA produces much better results than the RLA [49].”
- Cite reference(s) to support the assumption as well as (2) – (5) along with its notations, which was an issue not fully addressed regarding the original paper submission.
- “… (4)
where C is a constant independent…” (Remove the indentation.)
- “…to constrain the image and the kernel…”
- “… (7)
where α, γ are penalty parameters.” (Remove the indentation.)
- “…Just as the MIA [9], blind deconvolution is to minimize the loss function by obtain partial derivatives of J(o, h) with respect to o(x, y) and h(x, y), respectively, as follows, …”
- “Forcing (9) and (10) to zero, it will arrive at the maximum log-likelihood equations:”
- “…Multiply both sides of (11) and (12) by a positive actual number λ which is a parameter used to adjust the convergence rate of the algorithm.”
- “…Multiply both sides of (13) and (14) by the estimates of o(x, y) and h(x, y), respectively, to arrive at the final iterative formulae for image restoration.”
- “For (15) and (16), … in order to make the result converge and protect the edge information of the image while removing the noise, equations (15) and (16) can be rewritten as:…”
- Mathematical representations of the presented algorithm require further clarification and improvement.
While the manuscript cited [19] when formulating “sparsity prior to constrain the image and kernel”, it should define and clarify specific operation which the symbol “” performs in (5), and the rationale of using norm to define in (6) instead of norm as used in [19].
Clarify or revise the definition of “” in (9), (11),(13),(15) and (17). Based on the conventional notation on the left side of (9), “”, it seems that “” should be used to indicate that “” would be operated with respect to variable “” only. By the same token, “” should be used in (10), etc.
Clarify how “” and “” are computed or implemented exactly in the reported experiments.
Based on the description provided, the estimated blurred image at “th” iteration as shown in (17) was sharpened (or edge enhanced) using a Sobel vertical and horizontal edge detector and lowpass filtered by a Gaussian lowpass filter for noise reduction. The exact Gaussian lowpass filter implementation should be defined, and choice (or selection) of weight factor, , used for edge enhancement for be discussed or clarified. The mathematical representation of Sobel filtering process, in the form of seemed to be unconventional (i.e., mixing discrete convolution representation with matrix representation).
- “…It is the sigmoid function of the proposed BDA-SF that makes the critical difference from the MIA which uses the exponential function, and improves the blind deconvolution performance significantly. For comparison, these two functions are plotted here and shown in Figure 1. Figure 1(a) shows the plots of exponential functions with different coefficients, while Figure 1(b), a cluster of sigmoid functions. Figure 1 shows that the exponential function is asymmetric. …”
- In Tables 1-3, Figs.10-20, change “Ours” to “BDA-SF” as defined previously in the manuscript. Change “the proposed method” to “DBA-SF” or “the proposed DBA-SF”.
- “…The algorithm can effectively enhance the high frequency spectrum and achieve high-resolution restoration even when images are severely degraded. … The approach has added a low pass filter and edge-preserving process to the iteration formulae, to protect the image’s edges while removing noise sufficiently. For the image sequence, the method uses inter-frame information, by which satisfactory results can be obtained with fewer iterations. Extensive experiments demonstrate that the method achieves state-of-the-art for both natural images and images acquired under specific scenarios. It is expected that the success of deploying the sigmoid function in construction of the blind deblurring algorithm can motivate further research in the field of image restoration.”
- “[1] Jain, A.K. Fundamentals of Digital Image Processing. Prentice Hall 1989.”
Author Response
Thank you again for taking the time to review our manuscript.
Please see the attachment.

Reviewer 2 Report
I am happy with the authors response to the issues raised by the reviewers and the manuscript has significantly improved accomodating the reviewers suggestions. I support the acceptance of this manuscript for publication at the sensor journal.
Author Response
Thank you again for taking the time to review our manuscript.
Reviewer 3 Report
The authors have already addressed my key concerns. This work can be accepted.
Author Response
Thank you again for taking the time to review our manuscript.
This manuscript is a resubmission of an earlier submission. The following is a list of the peer review reports and author responses from that submission.
Round 1
Reviewer 1 Report
Dear authors
Reading his article: "Blind Deblurring Based on Sigmoid Function" was very helpful. The text is fluid and facilitates the understanding of your study.
I believe that the abstract does not adequately describe the quality of your work and should be improved in order to motivate the reading of the article.
Thank you if you consider the questions showed in the comments of the digital archive and the ones below:
1) The use of adjectives should be reviewed. For example, line 24: Image restoration algorithm in the spatial domain is the most famous. What would be the most famous?
2) Digital images have four levels of resolution: spatial, radiometric, spectral, and temporal. You refer to the level of detail present in the image so the resolution is spatial.
3) In equation 4 after multiplication by sigma squared, check if the denominator of the second term is correct.
4) Detail the process of the algorithm presented in lines 134 to 143.
5) Will the code be available on GitHub?
6) Visually, Figure 11b looks better than 11 am. You could comment more on that.
7) On line 203: We used evaluation indexes PSNR and SSIM to evaluate the image quality. Could you justify the choice of these two metrics considering the bibliographic references cited?
8) In table 2, method (24) is slightly superior to yours. How about commenting a little more about this result? The computational cost issue would be an advantage criterion.
I conclude by congratulating you on the quality of the work presented and wishing you a year of blessings and peace.
Sincerely

Reviewer 2 Report
The authors should provide a more detailed description of similar works.
The innovation of the proposed method in relation to the similar works should be more clearly presented.
The authors should describe with a more detailed pseudocode the whole algorithm.
The authors should better explain why the proposed methodology seems to work well.
In the conclusion, the authors should mention the limitations of their study and how these could be handled in a future work.
Reviewer 3 Report
This paper presented an image deblurring method based on the idea of multiplicative iterative algorithm for image sequences. The experiments show that the proposed method has better performance than the existing image deblurring methods.
However, the novelty is not enough. The proposed method is a clever implementation of existing handcrafted prior models, thus the paper should be better classified as a technical report. For example, why the authors used the dark channel prior on the model to constrain the kernel? My understanding of this is that the manuscript offers a minor tweak of [19]. Hence, the novelty of this work is ambiguous.
On the top of this, I think the authors should express more clearly which are the benefits of using multiple images from videos compared to deblurring from single images? This seems not realistic since five successive video frames would naturally lead to blurring artifacts due to dynamic scenes where parts of them are moving. The authors must discuss about this setting with respect to deblurring on videos.
It would be better to clearly state which the proposed method focuses on, grayscale or color images. I guess it is for grayscale images but it has not been described clearly.
I also wonder why we want to iterate over multiple images, not just a single image directly with the multiplicative iterative algorithm? Moreover, why using the sigmoid function? When you calculating the derivative of the sigmoid function, the gradients almost vanishing.
It is unclear to me whether the superiority of the proposed method over other deblurring methods comes from the sigmoid function, gradient sparsity prior, dark channel prior, or all of them. Lack of ablation study should be conducted in experiments.
I wonder if the experiments were a fair comparison. Why the proposed method that uses multiple images was only compared with singe image deblurring methods. If I am not losing something, the following video deblurring algorithms even seem to be a better competitor than those single image deblurring methods.
[39] Deep Video Deblurring for Hand-held Cameras, CVPR 2017
[r1] Video Deblurring Via 3d CNN and Fourier Accumulation Learning, ICASSP 2020.
Besides, there lacks ablation study that investigates the effectiveness of each hyperparameter. For example, what is the trade-off between the choice of hyperparameters?
Also, I suggest to the authors to consider the realistic VideoDeblurring dataset introduced in [39] since some samples in the used datasets look not blur but haze. This makes me not convincing about this paper.
In summary, I’m leaning toward negative.
Reviewer 4 Report
The manuscript presents and assesses a blind deblurring algorithm based on the sigmoid function.
Based in the experimental results, the authors claim that it demonstrates that the algorithm can remove the blur effect and improve the image quality on real and simulated images.
The methodology is clearly explained, the manuscript well organized and the results support the conclusions.
Despite the good quality of the research, I have the following recommendation to improve it:
- The abstract needs to be rewrited and stratified in an organized manner such as: Problem, motivation, methodology, main results, conclusions and future impact.
- Authors should avoid using personal pronouns such as "We" and "our", and replace it by impersonal writting.
- Only 18% of the references are of the last 5 years, more recent bibliography is required.
In summary:
- Rewrite the abstract
- replace the personal pronouns throughout the text by impersonal writing
- add more recent references, at least more 5 or 6, it is not difficult to find
Round 2
Reviewer 2 Report
The paper could be accepted in the current form
Author Response

(The authors gave the same response as above.)

Reviewer 3 Report
The revised version did not well address all my concerns. Although the authors have removed the use of dark channel prior, it still is unclear to me whether the superiority of the proposed method over other deblurring methods comes from the sigmoid function, gradient sparsity prior, or all of them. Lack of ablation study should be conducted in experiments.
In addition, the experiments in this paper are apparently not fair. The proposed deblurring method uses multiple images against the methods using only a single image, which means the proposed method has much more information to perform deblurring process in comparison with other single image based methods. This makes the contribution rather small.
A deblurring method working for only one dataset in a specific area is not a big contribution, it must be assumed that the method needs to be re-configured for each new dataset and area to be accurate. This makes me not convincing about this paper.
